# Dual-Modal In Vivo Fluorescence/Photoacoustic Microscopy Imaging of Inflammation Induced by GFP-Expressing Bacteria

**DOI:** 10.3390/s19020238

**Published:** 2019-01-10

**Authors:** Yubin Liu, Lei Fu, Mengze Xu, Jun Zheng, Zhen Yuan

**Affiliations:** Cancer Center, Faculty of Health Sciences, University of Macau, Macau SAR 999078, China; yubinliu@umac.mo (Y.L.); leifu@umac.mo (L.F.); mengzexu@umac.mo (M.X.); junzheng@umac.mo (J.Z.)

**Keywords:** GFP-expressing bacteria, biosensor, inflammation/infection, fluorescence imaging, photoacoustic microscopy

## Abstract

In this study, dual-modal fluorescence and photoacoustic microscopy was performed for noninvasive and functional in vivo imaging of inflammation induced by green fluorescent protein (GFP) transfected bacteria in mice ear. Our imaging results demonstrated that the multimodal imaging technique is able to monitor the tissue immunovascular responses to infections with molecular specificity. Our study also indicated that the combination of photoacoustic and fluorescence microscopy imaging can simultaneously track the biochemical changes including the bacterial distribution and morphological change of blood vessels in the biological tissues with high resolution and enhanced sensitivity. Consequently, the developed method paves a new avenue for improving the understanding of the pathology mechanism of inflammation.

## 1. Introduction

Inflammation denotes the immunovascular responses of biological tissues to harmful stimuli such as the pathogens, damaged cells, or irritants, which generally involves the immune cells, blood vessels, and molecular mediators [1]. Imaging of inflammation in vivo can pave a new avenue for improved understanding of the pathophysiology associated with various disease etiologies, such as heart disease, cancer, chronic respiratory disease, stroke, Alzheimer’s, diabetes, pneumonia, kidney disease, and flu [2,3,4,5,6,7,8,9,10,11]. Importantly, a number of non-invasive imaging techniques such as MRI, ultrasound, and CT, have been provided to study the biological mechanism of inflammation [12]. Among all imaging methods available, photoacoustic imaging (PAI) has exhibited its unique advantages in capturing both anatomy and function information of biological tissues with high optical contrast and high acoustic resolution [13,14,15,16,17,18,19,20]. PAI is a hybrid imaging technique that uses a pulsed laser to generate an acoustic wave. As the acoustic scattering is much lower than that from photonics, we can generate a high-resolution photoacoustic image. 

Present PAI has two major implementations: focused-scanning photoacoustic microscopy (PAM) and photoacoustic computed tomography (PACT). PAM aims to image millimeters deep at micrometer scale resolution, whereas PACT can be carried out to image centimeters deep at half-millimeter scale resolution. Typically, PAM enables the microcirculation/disease microenvironment studies of small animal models, revealing disease-induced abnormal microcirculations/vascular changes of biological tissues in vivo. [21,22,23]. By contrast, fluorescence imaging (FLI) is a very unique molecular imaging method, which has been extensively adopted as a powerful tool for inspecting cellular level events and cancer theranostics [24,25,26].

In this study, dual-modal fluorescence microscopy and photoacoustic microscopy imaging was presented as a high-sensitivity modality for inflammation detection and monitoring, which is able to take advantages of the complementary information from each technique. Single clones of GFP-expressing enteropathogenic *E. coli* were cultured in LB medium at 37 °C overnight. In addition, PAM and fluorescence microscopy imaging were performed based on the developed animal inflammation model after the injection of GFP-transfected bacteria. Consequently, fluorescence microscopy imaging can effectively map the diseased tissue with inflammation and track the distribution of bacteria in the biological tissues, whereas PAM can identify the in vivo immunovascular response including the vascular structure changes to inflammation. Since it is hard to distinguish and confirm the differences between the normal and inflamed tissue with one imaging modality, the developed dual-modal fluorescence and photoacoustic microscopy imaging approach definitely exhibits its unbeatable advantages in inspecting the inflammatory pathogenesis for various preclinical applications and clinical practices [27].

## 2. Methods and Materials

### 2.1. Preparation of GFP-transfected E. coli and In Vivo Animal Model

The GFP-transfected *E. coli* was cultured in a liquid Luria-Bertani (LB) medium (3 mL) at 37 °C overnight. The fluorescence intensity of GFP-transfected bacteria (or bacteria viability) after overnight cultured was quantified by microplate reader with excitation wavelength of 488 nm and emission wavelength at 525 nm. Next, the liquid LB medium was put on the orbital shaker incubator for 10–12 h, in which the shaking speed and temperature was set as 200 rpm and 37 °C, respectively. Further, the bacteria was washed with phosphate-buffered saline (PBS; Gibco, Thermo Fisher Scientific, Waltham, MA, USA) once and suspended in PBS. Finally, the bacteria was triply washed with PBS and then suspended in PBS to various concentrations for later imaging studies.

In addition, six eight-week old nude mice were used in this study. In vivo experiments were performed in compliance with the guidelines on animal research stipulated by the Animal Care and Use Committee at the University of Macau. After the mice were anesthetized with anesthesia machine (Isoflurane, R580, RWD Life Science, San Diego, CA, USA), the mice ear was imaged with our dual-modal photoacoustic and fluorescence microscopy systems before the bacteria infection. Furthermore, a 1 mL syringe with 30 g needle was utilized to inject the bacteria (concentration of 10^8^ cFU/mL, 50 µL) into the mice ear site to induce infection and later inflammation. Hereafter, the mice ear was imaged at 6 h post-injection by using the dual-modal imaging system.

### 2.2. The Dual-Modal Photoacoustic and Fluorescence Microscopy Imaging Systems

The homemade optical-resolution photoacoustic microscopy imaging system is provided in Figure 1, in which the pulsed laser (wavelength: 532 nm; pulse width: 1 ns; and repetition rate: up to 5 KHz) was used to illuminate the animals. The light beam first went through an optical subsystem (NDF: neutral density filter; CL1 and CL2: convex lens; PH: pinhole; OF: optical fiber), which was subsequently coupled with a single-mode fiber. Next, the laser beam delivered by the fiber changed its transportation direction by a rectangular prism (RP) after passing through an objective lens (OL1). To generate a high-resolution photoacoustic image, the second objective lens (OL2) was utilized, by which the laser beam focused on the imaging areas. A two-dimensional motor (M1 and M2) stage was rotated to scan the mice ear. The mice ear was fixed upon the agar gel phantom, the ultrasonic transducer was placed under the phantom, and a thin layer of ultrasound gel was spread over the interface between the agar phantom and transducer. The photoacoustic signals detected using a 50 MHz transducer with an element diameter of 5.08 mm and a focal length of 127 mm (VF 412, Valpey Fisher, Valpey Fisher, Hopkinton, MA, USA) were amplified by the amplifier (5073R, OLYMPUS, OLYMPUS CORPORATION, Waltham, MA, USA) and then collected on the computer to generate optical-resolution PAM images.

Specifically, the FLI system (eclipse Ni-U, Nikon Inc.) is a microscopy setup. As shown in Figure 2, this system consists of the mercury lamp, achromatic condenser, and CCD camera. For the present imaging system, the excitation wavelength is 488 nm and the emission filters used is 525~550 nm. In addition, the integration time (exposure time) is 100 ms, the objective magnification is 10 x, and the numerical aperture (N.A) is 0.3.

## 3. Results and Discussion

### 3.1. Relationship between the Various Concentrations of GFP-Expressing Bacteria and Fluorescence Signals

The fluorescence imaging capability of GFP-expressing bacteria with various concentration was examined; the results are provided in Figure 3. The fluorescence images regarding the medium filled with GFP-expressing bacteria with eight different individual concentrations (0 cFU/mL, 0.31 × 10^8^ cFU/mL, 0.62 × 10^8^ cFU/mL, 1.25 × 10^8^ cFU/mL, 2.5 × 10^8^ cFU/mL, 5 × 10^8^ cFU/mL, 10 × 10^8^ cFU/mL, and 20 × 10^8^ cFU/mL) was provided in Figure 3a. We discovered from Figure 3a,b that the fluorescence signals increased with increased concentrations of GFP-transfected *E. coli*. In contrast, as the GFP-expressing bacteria has the low optical absorption, the corresponding phantom tests were not performed based on PAM imaging, although PAM can capture the structure and function of high-absorption tissues, such as blood vessels.

### 3.2. Dual-Modal Imaging of Inflammation in Mice Ear

To inspect the GFP-expressing bacteria in vivo, the dual modal PAM and fluorescence microscopy was carried out by using the mice ear model for the investigation of inflammatory pathogenesis. The mice ear was injected with GFP-expressing bacteria and fluorescence and photoacoustic images were collected at pre- and 6 h post-infection. In particular, Figure 4a displays the picture of a normal mice ear, whereas Figure 4b,c shows the generated PAM and fluorescence microscopy images before the bacteria injection. Likewise, Figure 4d–f shows the mice ear with inflammation, the reconstructed PAM and fluorescence images at 6 h post-infection, respectively. In addition, the inflammation was monitored for 5 days based on the dual-modal imaging method and the imaging results are provided in Figure 5 for another mouse.

### 3.3. Discussion

We discovered from Figure 4 that the fluorescence signals at 6 h post-injection was significantly enhanced in the inflammation area, which exhibited dramatically increased bacterial populations in the infection region. The imaging results also demonstrated that fluorescence microscopy can effectively map the diseased tissue with inflammation and track the distribution of bacteria in the biological tissues.

Interestingly, we also discovered from Figure 5 that no significantly increased fluorescence signal was detected before the injection. However, enhanced fluorescence signal was observed and kept nearly unchanged at day one post-injection. More importantly, from the third day, the fluorescence signal intensity began to decrease; five days later, the signals totally disappeared. The dual-modal imaging results indicated that the developed system can be used to diagnosis and monitor bacteria-induced inflammation. However, it cannot capture the micro-environment/vascular changes of mice ear with inflammation due to the interaction between the tissues and bacteria. In addition, fluorescence imaging cannot identify the structure and function of normal tissues without infection.

By contrast, PAM can image the vascular structures of the normal mice ear. More importantly, as demonstrated in Figure 4e, inflammation with GFP-expressing bacteria in vivo can trigger strong immune response, which further result in increased blood vessels surrounding the infection site at 6 h post-infection (the oval represents the inflamed area). The results in Figure 5 also demonstrated that PAM imaging can effectively identify and monitor the vascular structures and micro-environment changes due to the inflammation.

## 4. Conclusions

In this study, homemade optical-resolution PAM systems combined with commercial fluorescence microscopy systems were used for imaging inflammation in mice ears. We discovered that mice ears with infections are a powerful animal model to study inflammatory pathogenesis. To date, it is still difficult to distinctively confirm the differences between normal and inflamed tissues by only using one imaging modality. As a result, dual-model imaging can be performed using PAM and fluorescence microscopy to improve the sensitivity for the detection and monitoring of inflammation.

## Figures and Tables

**Figure 1 sensors-19-00238-f001:**
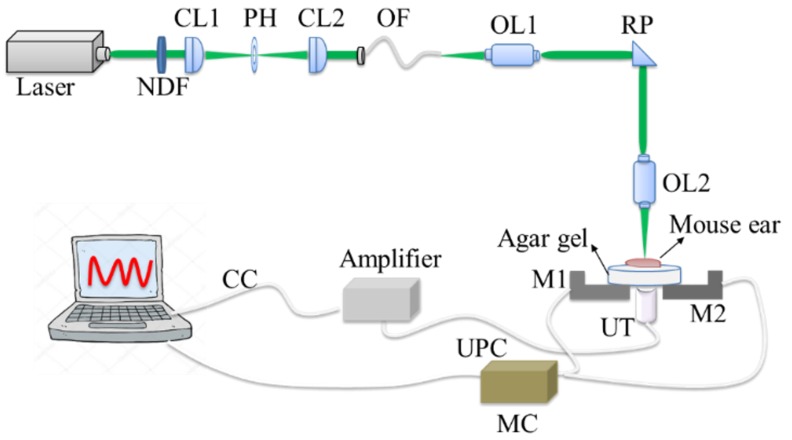
The home-made optical-resolution photoacoustic microscopy (PAM) imaging system. NDF: Neutral density filter; PH: pinhole; CL1 and CL2: Convex lens; OF: Optical fiber; OL1 and OL2: Objective lens; RP: Rectangular prism; M1 and M2: Two-dimensional motor; MC: Motor Control; UT: Ultrasonic transducer; UPC: Ultrasonic probe cable; and CC: Coaxial cable.

**Figure 2 sensors-19-00238-f002:**
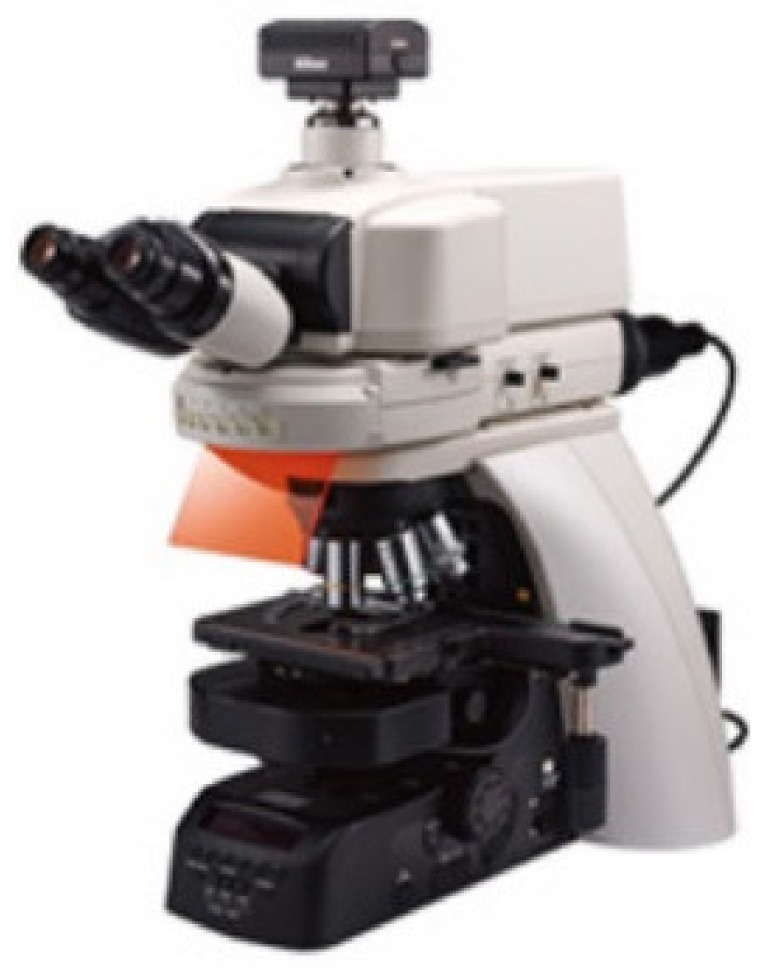
The commercial fluorescence microscopy imaging system.

**Figure 3 sensors-19-00238-f003:**
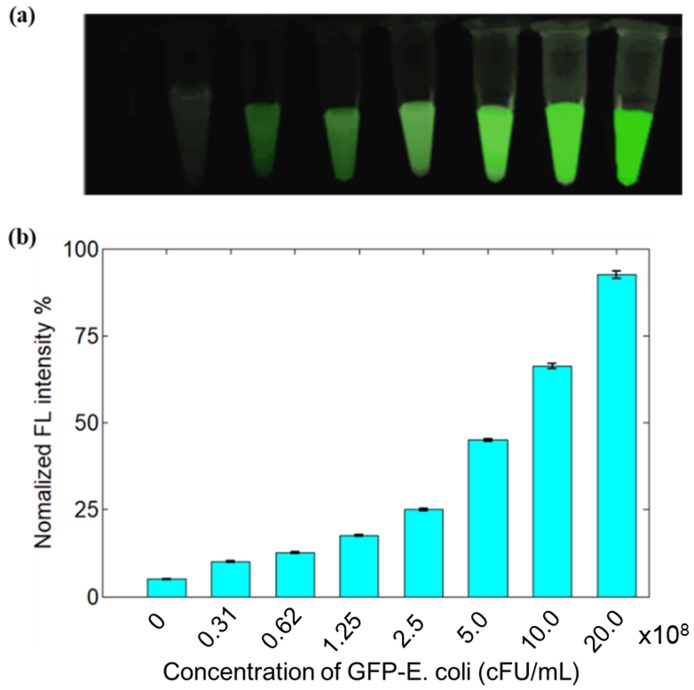
Fluorescence signals compared to different concentrations of green fluorescent protein (GFP)-transfected *E. coli* (**a**) and the relationship between the fluorescence intensity and various concentrations of GFP-transfected *E. coli* (**b**). cFU: Colony-Forming Units.

**Figure 4 sensors-19-00238-f004:**
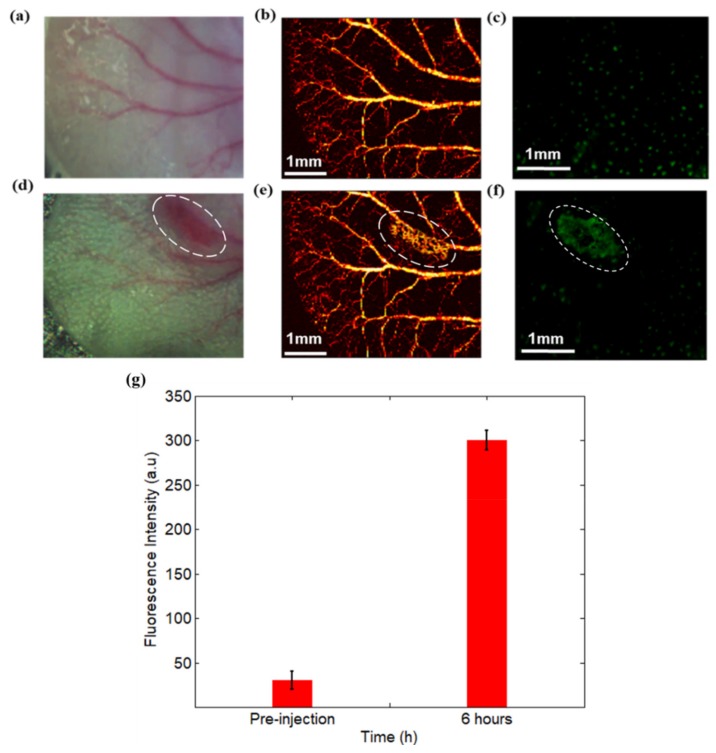
Dual-modal PAM and fluorescence microscopy imaging results for the first mouse. (**a**) The photography of normal mice ear; (**b**) PAM imaging of normal mice ear; (**c**) fluorescence microscopy imaging of normal mice ear; (**d**) photography of the same mice ear at 6 h post-injection; (**e**) PAM imaging of immunovascular response of mice ear with inflammation at 6 h post-injection; (**f**) fluorescence microscopy imaging of the bacteria distribution and inflammation regions of mice ear at 6 h post-injection; and (**g**) the corresponding fluorescence intensity at pre-injection and 6 h post-injection.

**Figure 5 sensors-19-00238-f005:**
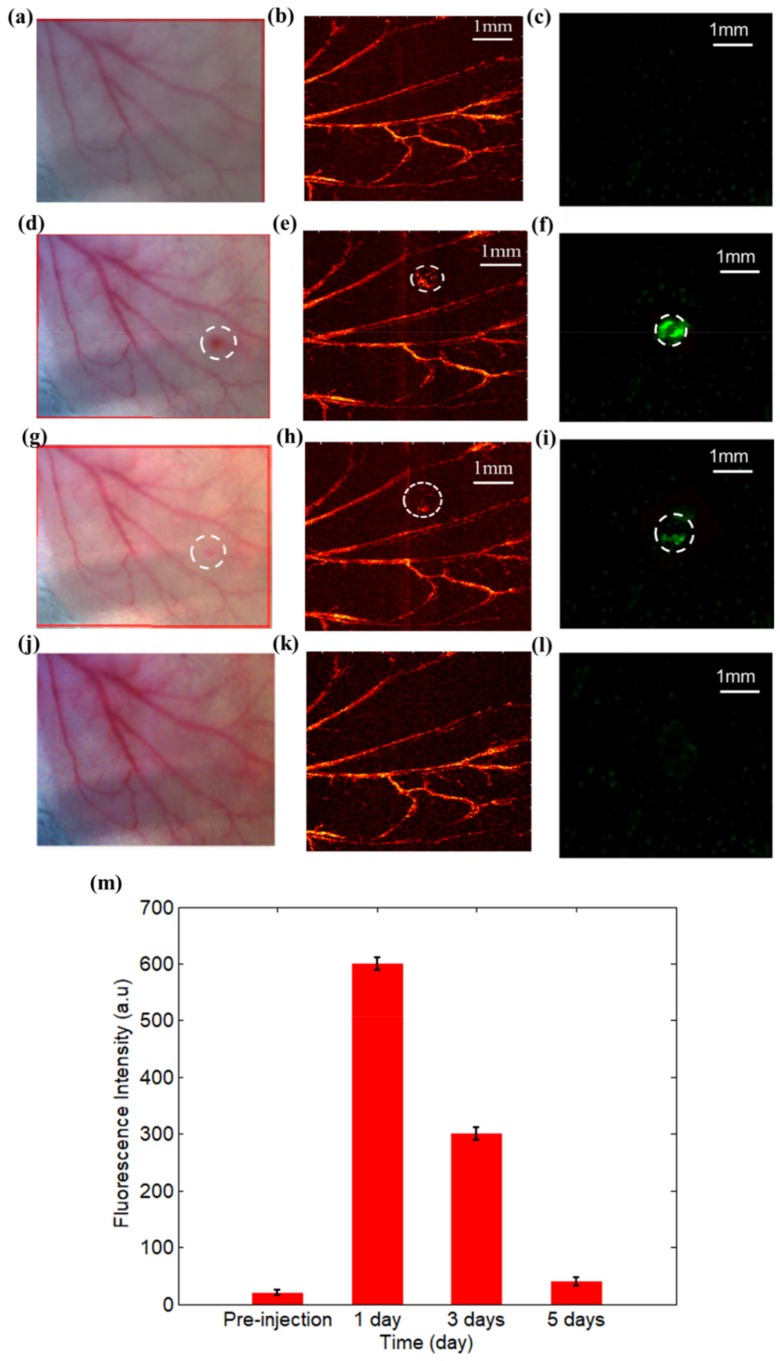
Dual-modal PAM and fluorescence microscopy imaging for monitoring the inflammation: (**a**–**c**) are the imaging results before the injection of bacteria; (**d**–**f**) are the results at 1 day post-injection; (**g**–**i**) are the results at 3 days post-injection; (**j**–**l**) are the results at 5 days post-injection. The first column showed the picture of mice ear, the second one displayed the photoacosutic image, whereas the third one exhibited the fluorescence image; (**m**) the corresponding fluorescence intensity at pre-injection, 1 day, 3 days, and 5 days post-injection, respectively.

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
