# Peer review of "Dual-Modal In Vivo Fluorescence/Photoacoustic Microscopy Imaging of Inflammation Induced by GFP-Expressing Bacteria"

_sensors, 2019, doi:10.3390/s19020238_

Round 1

Reviewer 1 Report

I appreciate the paper as it may have to be interesting to scientists in the field and potentially show a way of modern diagnostics. I have several minor issues to discuss and/or be corrected:

line 69: Means of ansthetization is not specified.

line 111: There were eight different concentrations of GFP-expressing bacteria solutions used in the experiments. I would expect explanation of 1. choosing the maximum 20 x10^8 cFU/ml, 2. choosing of exponential differences between individual concentrations.

Author Response

Reviewer 1

Comments and Suggestions for Authors

I appreciate the paper as it may have to be interesting to scientists in the field and potentially show a way of modern diagnostics. I have several minor issues to discuss and/or be corrected:

We thank the reviewer for your constructed comments and encouragement.

line 69: Means of ansthetization is not specified.

We are sorry for that. As suggested, the information on anesthetization was added in the revision (see lines 71 and 72 on page 2).

line 111: There were eight different concentrations of GFP-expressing bacteria solutions used in the experiments. I would expect explanation of 1. choosing the maximum 20 x10^8 cFU/ml, 2. choosing of exponential differences between individual concentrations.

We are sorry that we didn’t state clearly on the concentrations for the used GFP-expressing bacteria solutions. Now, it (concentration of 108 cFU/ml, 50µL) was added in the revised manuscript (see line 75 on page 2).

In fact, the second largest concentration was utilized in this study to generate the strong fluorescence signal for in vivo experiment.

Reviewer 2 Report

Liu et al presented an imaging based study on E. coli infection induced inflammation of mouse ear employing photoacoustic microscopy as well as fluorescence microscopy. The manuscript could gain from the following improvements.

1.      Section 1, line 37, please elaborate what is meant by “high ultrasound resolution”.

2.      Page 2, line 53- 53, what is meant by “hard to distinctive”? Perhaps the author means distinguish? Also change “inflammation tissues” to inflamed tissue

3.      In the introduction section, discuss previously reported dual mode photoacoustic imaging and fluorescence imaging (for example 10.1364/BOE.7.002692). In addition, a brief description on the techniques will be helpful for the readers.

4.      The introduction should end with a summary of the experiments described in the manuscript.

5.      Define all abbreviations before first use example, GFP, LB and so on.

6.      Page 2, line 62, please correct “anesthetized with **,”

7.      Page 4 , line 129, what is meant by “mice ear modal”?

8.      What is the purpose of Figure 2? Since it is a commercial instrument, it does not add anything to the manuscript. Moreover, the authors should provide fluorescence imaging parameter like excitation wavelength, emission filters used, integration time, and objective magnification and so on.

9.      Please elaborate on PAM image processing algorithms used.

10.   Page 4 line,123-124, “mice ear with inflammation”, how was the presence of inflammation confirmed?

11.   Why did the author look at 6 hours timepoint? Does this duration have any significance in bacteria related infection?

12.   Is the mice ear still viable after 6 hours from excision?

13.   Section 3.3, line 128, quantify the increase if fluorescence signal before and 6 hours post –injection. Is this increase statistically significant?

14.   How many mice ( sample size) were used for this experiment? Did the author use only the images shown to draw their conclusion? Sufficient samples should be used to confirm the observations

15.   Section 3.3, line 3.2, it is stated that they were able to “track the distribution of bacteria in the biological tissues”. However, no quantitative data has been provided to support this claim. Just image4(f) is not sufficient to track distribution.

16.   Can the PAM results be quantified?

17.   Since the PAM images and fluorescence images are not in the same scale, it is difficult to understand the relative spatial location of the inflammation and the bacteria ( green region). It would be helpful if this could be made clear for the readers.

18.   Page 4, line 140-141. Please elaborate how could the PAM images indicate “micro-environment changes due to the inflammation” Can this be quantified?

Author Response

Reviewer 2

Comments and Suggestions for Authors

1.      Section 1, line 37, please elaborate what is meant by “high ultrasound resolution”.

In the revision, the “high ultrasound resolution” was changed to “high acoustic resolution” to avoid misleading.

Photoacoustic imaging (PAI) is a hybrid imaging technique, which uses pulsed laser to generate acoustic wave. As the acoustic scattering is much lower than that from photonics, we can generate a high-resolution PA image.

The new descriptions were added in the revision (see lines 33 to line 36 on page 1).

2.      Page 2, line 53- 53, what is meant by “hard to distinctive”? Perhaps the author means distinguish? Also change “inflammation tissues” to inflamed tissue

We thank the reviewer for pointing out our typo errors. As suggested, the errors were corrected accordingly.

3.      In the introduction section, discuss previously reported dual mode photoacoustic imaging and fluorescence imaging (for example 10.1364/BOE.7.002692). In addition, a brief description on the techniques will be helpful for the readers.

The previous reference was added in the revision (see reference *). The previous work was mainly focused on in vitro study. By contrast, our novelty lies in on two perspectives: 1) dual-modal in vivo fluorescence/photoacoustic microscopy imaging was utilized in this study; 2) GFP-transfected E. coli was used to generate fluorescence and tracking inflammation.

4.      The introduction should end with a summary of the experiments described in the manuscript.

As suggested, it was added (see line 48 to line 51 on page 2).

5.      Define all abbreviations before first use example, GFP, LB and so on.

As suggested, they were defined before first use.

6.      Page 2, line 62, please correct “anesthetized with **,”

Added in the revision (please (see lines 71 and 72 on page 2).

7.      Page 4 , line 121, what is meant by “mice ear modal”?

In the revision, the “mice ear modal” was changed to “mice ear model”.

8.      What is the purpose of Figure 2? Since it is a commercial instrument, it does not add anything to the manuscript. Moreover, the authors should provide fluorescence imaging parameter like excitation wavelength, emission filters used, integration time, and objective magnification and so on.

As suggested, the fluorescence imaging parameters were added in the revision.

The excitation wavelength is 488nm and the emission filters used is 525~550nm. The integration time (exposure time) is 100ms, the objective magnification is 10x, and the numerical aperture (N.A) is 0.3.

These were added in the revision (see lines 100 to 105 on page 3).

9.      Please elaborate on PAM image processing algorithms used.

In this study, the PAM images were directly generated from the acquired A-scanned acoustic signals. We used a two-dimensional platform to move the sample, and then the peak value of photoacoustic signal from each scanning point was used to recover the images.

10.   Page 4 line,123-124, “mice ear with inflammation”, how was the presence of inflammation confirmed?

The inflammation was caused by the bacteria infection which can be confirmed by the fluorescence signal generated from growth of GFP-transfected bacteria (only bacteria can generate fluorescence signals).

11.   Why did the author look at 6 hours timepoint? Does this duration have any significance in bacteria related infection?

For our experiments, we discovered that the fluorescence signal was significantly enhanced between 6 hours to 1 day post-injection. The duration has significance in bacteria related infection since the growth rate of GFP-transfected bacteria was significantly increased and fluorescence signal was also very strong during this period.

12.   Is the mice ear still viable after 6 hours from excision?

The mouse was still alive after 6 hours in vivo tests. We generally monitored the mice ear for 5 days before the mice were sacrificed.

13.   Section 3.3, line 128, quantify the increase if fluorescence signal before and 6 hours post –injection. Is this increase statistically significant?

Interestingly, we didn’t observe any significantly increased fluorescence signal at 4 hours post-injection. However, at 6 hours post-injections, enhanced fluorescence signal was detected and was kept nearly unchanged for one day. From the second day, the fluorescence signal intensity began to decrease and five days later, the signals totally disappeared. The new results for monitoring were provided in the revision (see new Figure 5).

14.   How many mice (sample size) were used for this experiment? Did the author use only the images shown to draw their conclusion? Sufficient samples should be used to confirm the observations

In this study, six mice were used for the experiments and the observations were made based on the mean signals from all mice.

15.   Section 3.3, line 3.2, it is stated that they were able to “track the distribution of bacteria in the biological tissues”. However, no quantitative data has been provided to support this claim. Just image4(f) is not sufficient to track distribution.

In this study, the bacteria were transfected by GFP and only bacteria can produce the fluorescence signals. As such, we can track the distribution of bacteria in the biological tissues.

16.   Can the PAM results be quantified?

Certainly the PAM images can be quantified with measurements from multiple wavelength. Presently, we can only identify the structure information with relative optical absorption based on the measurements from one wavelength. Further work should be performed to capture the function information.

17.   Since the PAM images and fluorescence images are not in the same scale, it is difficult to understand the relative spatial location of the inflammation and the bacteria (green region). It would be helpful if this could be made clear for the readers.

In the revision, the spatial information was modified to match each other between the the PAM images and fluorescence images (please see new figure 4).

18.   Page 4, line 140-141. Please elaborate how could the PAM images indicate “micro-environment changes due to the inflammation” Can this be quantified?

The blood vessel and blood oxygen changes at microenvironment can be monitored by PAM with multiple-wavelength measurements. This should be able to be quantified and further work should be conducted to resolve this issue.

Round 2

Reviewer 2 Report

The response is satisfactory. However, in response to question 13, the authors still did not provide quantification of fluorescence intensity data to show whether increase of fluorescence signal was statistically significant. 

Author Response

Comments and Suggestions for Authors

The response is satisfactory. However, in response to question 13, the authors still did not provide quantification of fluorescence intensity data to show whether increase of fluorescence signal was statistically significant.

Thanks for the reviewer’s comments. The quantification of fluorescence intensity results are added in figure 4 and figure 5.
